# The Role of Social Media in Internalizing Body Knowledge—A Cross-Sectional Study among Women with Different Food Preferences

**DOI:** 10.3390/ijerph20032069

**Published:** 2023-01-23

**Authors:** Karolina Krupa-Kotara, Mateusz Grajek, Mateusz Rozmiarek, Ewa Malchrowicz-Mośko, Wiktoria Staśkiewicz, Patxi León-Guereño, Aitor Martínez Aguirre-Betolaza, Arkaitz Castañeda-Babarro

**Affiliations:** 1Department of Epidemiology, Faculty of Health Sciences in Bytom, Medical University of Silesia in Katowice, 41-902 Bytom, Poland; 2Department of Public Health, Department of Public Health Policy, Faculty of Health Sciences in Bytom, Medical University of Silesia in Katowice, 41-902 Bytom, Poland; 3Department of Sports Tourism, Faculty of Physical Culture Sciences, Poznan University of Physical Education, 61-871 Poznan, Poland; 4Department of Food Technology and Quality Evaluation, Department of Human Nutrition, Faculty of Health Sciences in Bytom, Medical University of Silesia in Katowice, 41-808 Zabrze, Poland; 5Department of Physical Activity and Sports, Faculty of Education and Sport, University of Deusto, 48-007 San Sebastian, Spain; 6Department of Physical Activity and Sports, Faculty of Education and Sport, University of Deusto, 48-007 Bilbao, Spain

**Keywords:** social media, traditional diet, vegetarian diet, internalization, self-esteem, body image

## Abstract

Virtual spaces, such as social media, give people a platform to exchange their opinions, experiences, and knowledge. Social media’s ubiquitous usefulness has led to people relying, in whole or in part, on the information they learn online. As a result, a person’s perception of his or her own body and their self-worth has started to be influenced by what other people think of them and by the information found on social media. Modern people’s lifestyle, particularly their eating habits and exercise habits, exhibits a similar tendency. The purpose of this study was to evaluate the relationships occurring between factors related to the use and internalization of body image knowledge contained in social media and the factors shaping self-assessment and self-esteem in women following a vegetarian diet. An authority-validated questionnaire was used to determine the level of use and attitudes of respondents toward social media, e.g., SATAQ and BES. Associations about the potential negative impacts of the knowledge provided by social media on the development of body image and self-esteem were shown. It is interesting to observe that women who practice vegetarianism have less pressure on their appearance. This may be because vegetarianism is a lifestyle that is currently actively promoted online. Education is required on the connection between the use and internalization of social media knowledge and the development of healthy self-esteem and body evaluation.

## 1. Introduction

Social media is a vital component of everyone’s life in the modern era. It serves as the foundation for the way we communicate, exchange ideas and content, and build stronger communities among professionals. Social media can be categorized based on how it is used. These include applications for expressing opinions, building friendships, communicating, cooperating, or sharing resources [1,2]. As such, social media has two main roles. Informational—consisting of acquiring, expanding, and sharing knowledge gained online, and social—based on building relationships in the virtual space. Facilitated access to and increased use of technology has made social media a permanent part of life, especially for today’s youth, referred to as Generation Z [2,3]. As a result, social media has an effect on many aspects of their lives. They affect the user’s physical and emotional health in addition to online communication, learning, and interpersonal interaction [1,2,3]. They are mostly in charge of creating dietary preferences and trends, such as diets that exclude meat and zoonotic foods.

The public adoption of vegetarian and vegan diets has a connection to health issues as well. Unfortunately, whether vegan and vegetarian diets have a favorable or detrimental impact on health cannot be shown. This has to do with the fact that vegetarians and vegans do not make up a single demographic. Depending on the type of diet they select, how they make up for the absence of meat and zoonotic products in their diet, and how long they have been on a meat-free diet, they will differ. Therefore, the homogeneity of the study group should be examined in any study that discusses the advantages or disadvantages of a vegetarian or vegan diet [4,5]. One can learn about the benefits of vegetarian and vegan diets on body mass index (BMI), blood glucose and cholesterol levels, and the risk of developing cancer and cardiovascular disease by reading scientific studies [5].

Social media, which is a virtual environment, offers a forum for the expression of ideas, the sharing of personal stories, and the gathering of knowledge about important topics. In Poland, a poll of social media users’ motivations found that social connection was the most frequently cited reason for using social media [6]. Other studies show that in addition to one’s persona on social media, there is also a presentation of one’s lifestyle, actions, and interests, which could include things, such as narcissism or the desire to advance one’s status, for example [7]. This can be accomplished, among other methods, by engaging in athletics or volunteering [8]. Social media platforms do really offer a wonderful setting for attracting attention and recognition. People now rely largely or entirely on knowledge learned online due to social media’s broad functionality [9]. As a result, a person’s perception of his or her own body or sense of self-worth has started to be influenced by what other people think about them and via the lens of information found on social media. Similar circumstances exist.

Accordingly, the main objective of the study was to assess the relationships occurring between factors related to the use and internalization of body image knowledge contained in social media and factors shaping body evaluation and self-esteem in women following a vegetarian diet. The main objective was also to assess the differences occurring in the aforementioned factors between those following a vegetarian diet and those following a traditional dietary model. By the main objective of the study, two research questions were posed: (1) Does internalizing social media knowledge affect the negative self-esteem of the women surveyed?; (2) Do the surveyed women who follow a vegetarian diet show less impact of internalizing body image knowledge from social media?

## 2. Materials and Methods

### 2.1. Study Area and Research Sample

The survey was conducted using the CAWI (computer-assisted web interview) method. A proprietary questionnaire, along with psychometric tests, was posted on the Google Forms platform. The form was distributed using social media, such as Facebook and Instagram. Responses were collected between 1 September 2021 and 30 January 2022. The survey was fully anonymous and voluntary, which respondents were informed of in the initial instructions for the survey. Respondents could opt out of the survey at any time.

The online survey method was chosen as the most appropriate for conducting a study on the topic of the cyberspace environment (social media) [10]. To minimize biased questionnaire completion and bias, preventive tools were used in the form of checking the time of logging into the form (completion times of less than 5 min were rejected), and questionnaires completed from the same IP address were handled similarly. In addition, CAPTCHA keys and anti-fake/bot responders tools were used.

A total of 520 women participated in the study. Sixty of them were excluded from the study due to faulty completion of questionnaires and the lack of research tools to analyze questionnaires among non-binary people. In the end, 460 people qualified for the survey. Due to poor questionnaire maneuverability in the male population, the study was abandoned in this group and focused only on the female population.

The necessary sample size was calculated according to the population size of the Polish region. It was estimated that a sample of 466 people would be sufficient and representative of Poland. It was assumed, according to the CSO report [11], that the population of Poland is 38,036,100 people. The sample size was calculated according to the formula: Nmin = NP ⋅ (α^2^ ⋅ f(1 − f)) ÷ NP ⋅ e^2^ + α^2^ ⋅ f(1 − f), where: Nmin—minimum sample size, NP—the size of the population from which the sample is drawn, α—confidence level for the results, f—the size of the fraction, e—assumed maximum error. For the Polish population, the minimum sample size of respondents was calculated, which was 384 people (α = 0.95; f = 0.5; e = 0.05). Based on these calculations, the collected group was considered representative. No significant statistical differences (*p* < 0.05) were found between the participants in the study, so it was assumed that the group represents a homogeneous population in terms of the characteristics presented.

### 2.2. Inclusion Criteria for the Study

The sample was selected based on the following criteria: voluntary consent of survey participants and being 18 years of age. Respondents who did not meet the above criteria were not qualified for the study. The study complies with the provisions of the Declaration of Helsinki, as amended. In addition, the work plan included conducting a pilot study and calculating the Kappa parameter. The study was approved by the Bioethics Committee of the Silesian Medical University in Katowice (ID. PCN/CBN/0052/KB/127/22).

### 2.3. Study Procedure and Research Tool

The survey was conducted in three stages. The first stage was a pilot study, during which 30 randomly selected respondents were asked to fill out a questionnaire to check that all questions were understood. The majority of questions were found to be clear and understandable to respondents, while questions that were indicated by at least 2 respondents as incomprehensible or unclear were removed or corrected. Stage two was the validation of the questionnaires by distributing them twice to a randomly selected group of 100 respondents. An interval of 2 weeks was maintained between co-locations of the questionnaires. The consistency of responses to the same questions was checked. To assess the reproducibility of the results obtained by the questionnaire used, the value of the κ (Kappa) parameter was calculated for each question in the questionnaire; for 63.3% of the questions, a very good (κ ≥ 0.80) method consistency was obtained, while for 36.7% of the questions, a good (0.79 ≥ κ ≥ 0.60) method consistency was obtained. The final stage of the study was to conduct an active test.

The survey questionnaire was used to obtain information about the socioeconomic status of the respondents and to determine the level of use and attitude of the respondents toward social media.

The Sociocultural Attitudes Toward Body Appearance Questionnaire, SATAQ 3, by Izydorczyk and Lizinczyk, is used for the subjective evaluation by the respondent of the influence of social media, TV, and magazines on the formation of attitudes toward the body and appearance. The questionnaire consists of 28 items relating to, among other things, image pressure created by mass media, e.g., “I feel pressure coming from TV and magazines to lose weight”. The respondent, using a score from 1–5, in which 1 means “strongly disagree” and 5 means “strongly agree”, determines the degree of truthfulness of the items given. The responses obtained were classified according to four scales: pressure internalization, i.e., feeling pressure to conform one’s appearance and image to the norms flowing from social media; information seeking internalization, i.e., using social media to obtain information on the standards of promoted body image and implement actions to change one’s current appearance; internalization athleticism, i.e., the need to have an athletic body and take actions to achieve it; and information on the frequency of searching mass media for standards of promoted body image and appearance. The test is characterized by satisfactory psychometric properties. Cronbach’s internal consistency coefficient α for individual scales ranged from 0.76 to 0.92 [12].

Respondents’ attitudes toward their bodies were examined using a Polish adaptation of the Franzoi and Shields Body Esteem Scale (BES). The tool consists of 35 items divided into 3 subscales, differentiated by gender. The items are sexual attractiveness, relating to body parts whose emphasis, such as through makeup, enhances a woman’s sexuality; weight control, relating to body parts subject to change through exercise; and physical fitness—used to assess the level of satisfaction with the strength or agility of the body. The subject assesses their attitude toward each item on a 5-point Likert-type scale, in which 1 means “I have strong negative feelings” and 5 means “I have strong positive feelings”. The BES test has satisfactory psychometric properties. Reliability for scales designed for women ranged in Cronbach’s α between 0.80 and 0.89 [13].

The final psychometric questionnaire is the Rosenberg Self-Esteem Scale. The study used a Polish adaptation of the questionnaire by Dzwonkowska, Lachowicz-Tabaczek, and Laguna. The scale contains 10 items relating to beliefs about oneself, e.g., “I like myself” or “I believe I have many positive qualities”. The respondent, using a score from 1 to 4, in which 1 means “strongly agree” and 4 means “strongly disagree”, determines the truthfulness of the items given. The level of self-assessment is determined by the total score obtained. The SES test has satisfactory psychometric properties. Cronbach’s internal consistency coefficient α for different age groups ranged from 0.81 to 0.83 [14].

### 2.4. Statistical Analysis

Statistical processing of the results was carried out using STATISTICA version 13.3. The normality of the distributions was assessed using the Shapiro–Wilk test and the Kolmogorov–Smirnov test. Non-parametric evaluation of relationships between variables was used, creating correlation matrices using Spearman’s rank correlation coefficient. Intergroup comparisons were made using the nonparametric Mann–Whitney U test. A statistical significance level of *p* ≤ 0.05 was used for statistical analyses. In statistical testing, corrections were applied to obtain more accurate calculations.

## 3. Results

Depending on the sort of food each subject consumed, the participants were divided. There were two groups: one that had a vegetarian diet (plant-based), and the other that followed a conventional dietary model in contrast (traditional). The first group included 200 people following a vegetarian diet (vegetarian or vegan)—aged 18–68 years (M = 28.63 ± SD = 9.34). The comparison group consisted of 260 people following a traditional dietary model—aged 18–68 (M = 30.22 ± SD = 9.25). The basic socio-demographic characteristics of the group following a vegetarian diet are shown in Table 1.

In women following a vegetarian diet, statistically significant associations were observed between factors related to the use and internalization of body image information from social media and age, BMI, self-esteem, and attitude toward one’s own body.

The pressure internalization factor was found to be significantly and negatively associated with the age of the subjects, as well as with sexual attractiveness, weight control, focusing on physical activity, and self-esteem. The positive associations of pressure internalization were observed about the intensity of the need to have a muscular figure, information seeking, and the frequency of seeking information on body image and appearance. The intensity of the ability to have a muscular physique was also positively related to the search for information on body image and the frequency of such searches. Similarly, seeking information on body image significantly and positively correlated with the frequency of seeking information on social media. The negative associations of seeking information on body image were noted for age, sexual attractiveness, weight control, focusing on physical activity, and self-esteem. The frequency of seeking information on social media about one’s body image also correlated negatively with age, sexual attractiveness, weight control, and focusing on physical activity and self-esteem. Self-esteem factors (sexual attractiveness, weight control, and focusing on physical activity) are strongly and positively correlated with each other. They also significantly and positively correlated with self-esteem. It is also worth mentioning that BMI showed negative associations with weight control and focusing on physical activity. In general, the use of social media and the internalization of body image patterns from social media were negatively associated with self-esteem and self-assessment. The strength of the associations is presented in Table 2.

Statistically significant differences were found between women following a vegetarian diet and those following a traditional dietary model in terms of factors, such as internalization pressure, internalization of the information searched, frequency of information searched. Higher intensities of these factors were observed in women following the traditional dietary model (Table 3).

## 4. Discussion

The research we performed enabled us to get confirmation of the established hypotheses. It has been demonstrated that among vegetarian women, using social media and internalizing knowledge about body image are highly related to feeling pressure about one’s looks. The frequency of obtaining such information on social media is rising along with feelings of heightened pressure regarding one’s physical appearance, and this has a detrimental impact on respondents’ opinions of their bodies. This may be due to the fact that the recipients of offered content rarely exhibit the “ideal” body patterns that are displayed in social media. There is a phenomena where people compare their own bodies to those in social media as a result.

Similar results were obtained in a review conducted by Saiphoo and Vahedi [15]. They showed a significant and positive relationship between the use of social media and disturbances in the evaluation of one’s own body. This phenomenon was observed both among users who believe in the veracity of the content presented on social media, as well as among users who are aware of the modifications to the content shared on social media [16]. Authors Franchina and Lo Coco came to similar conclusions, noting lower satisfaction with their bodies among social media users [17]. They explained this phenomenon by increased exposure to photos and posts depicting the “ideal” body among social media users. The authors of the aforementioned articles also pointed out that women are characterized by higher pressure and internalization of body image knowledge contained in social media, and this, in turn, affects their level of self-esteem. The same results were obtained in the present study. This is because it was proven that feeling pressure from social media regarding body image increases the frequency of seeking information about the “ideal” figure. This, in turn, increases efforts to conform one’s appearance to the pattern found on the Internet while decreasing satisfaction and evaluation of one’s own body among women. As the factors responsible for evaluating one’s own body (sexual attractiveness, weight control, attention to physical fitness) positively correlated with self-esteem, dissatisfaction with one’s appearance may be negatively associated with a woman’s self-esteem. The use of social media can cause a disturbance in women’s perception of their bodies, leading to a “gap between the ideal body and one’s own”. The problem is the phenomenon of objectification of women in culture, whereby there is a prioritization of body appearance rather than intelligence or manner. As a result, a woman seeks to increase her self-worth by conforming the appearance of her own body to the prevailing standards presented in the mass media [18]. Similar findings were made in a study on the impact of modifying social media images on women’s self-esteem and happiness with their own bodies. It was discovered that simply posting a photo of oneself on social media, whether it had been modified or not, was linked to emotions, including worry, dread, low self-esteem, and body image dissatisfaction. Additionally, allowing the study’s participants to modify published pictures led them to highlight flaws in their own bodies, which lowered their self-esteem [19].

In their systematic review, Revranche and colleagues [20] emphasize that there is strong support in the literature for the association between using social networks and having a poor perception of one’s body. A nested measure of social network use, such as particular activities featuring appearance-related content, may be used to predict negative body image instead of research examining the frequency of overall use. Future research might instead evaluate behaviors that are typically present on a certain platform in light of the disparities between the self-reported frequency of social network use and actual use noted in the methodological literature. In addition, there is a need to distinguish specific categories of sites, such as highly visual social media, when focusing on body image results. Focusing on specific social media platforms may, in turn, lead to more targeted prevention regarding safe use of social networking sites among teens. Despite the growing body of research on the relationship between social media and body image, the review emphasizes that additional longitudinal and experimental studies are needed to explore potential bidirectional effects, as well as studies based on representative samples to improve generalizability to adolescent populations. Jarman [21] and Kimmerle [22] confirm this in their studies.

It is challenging to locate research addressing variations in self-evaluation and self-esteem between those following a vegetarian or vegan diet and those following a traditional dietary model after reviewing the Polish and foreign literature. However, the topic of quality of life among vegetarians, which encompasses psychological, physical, social, and spiritual dimensions, as well as overall wellbeing, warrants further investigation [23,24]. Indeed, it has been shown that people following a vegetarian and vegan diet have a better quality of life compared to those following a traditional dietary model [25,26]. The main reasons for the quality of life in vegans and vegetarians are the overall improvement in physical health, positive emotions associated with adopting a lifestyle that is beneficial to health and the environment, and a sense of belonging to a certain social group. These factors can positively correlate with self-esteem and self-evaluation. In addition, there are an increasing number of posts and articles on social media about the need to change lifestyles and, most importantly, reduce meat consumption to reduce the negative effects of food production on the climate [26]. Thus, those who follow a vegetarian diet that is socially accepted and promoted on social media will experience higher levels of self-satisfaction, which may ultimately translate into self-esteem [27,28]. A study by Gwioździk et al. [28] found that women on a traditional diet had the highest scores for uncontrolled and emotional eating. Women on a vegetarian diet, on the other hand, had the lowest scores for uncontrolled and emotional eating of all subjects. Results supporting the above claims were obtained in the groups of women on a vegetarian diet in this study. Indeed, significant differences were found in the level of pressure and internalization of self-image information between women following a vegetarian diet and those following a traditional dietary model. The group following the traditional dietary model was characterized by higher pressure and internalization, as well as an increased frequency of seeking body image information on social media. As mentioned earlier, this may be because, currently, vegetarian dieters are heavily promoted in the virtual space, and as a result, female respondents already following this type of diet may not feel significant pressure to change their image or lifestyle compared to those following the traditional dietary model. It should be taken into account that there are also studies indicating the reduced quality of life and well-being among those following a vegetarian diet compared to those following a traditional dietary model [28,29,30,31,32,33]. Thus, there is a need to continue and expand research in the area of self-esteem, self-assessment, and the quality of life among people following a vegetarian diet.

## 5. Strengths and Limitations

The group selection adopted in the survey helped reduce the risk of bias, and conducting a pilot survey and calculating kappa statistics also increased its cognitive value. The survey conducted is not free of methodological limitations, which will be excluded in future projects. Conducting the research using the indirect online survey method (web forms) does not avoid the common “bot/fake respondents” phenomenon that characterizes surveys shared via social media and common forums, but the nature of the survey and its scope justifies the methodology adopted. In further research, the group of respondents should be expanded to include men. It would be worthwhile to continue research in the population of young people, who do not currently have the opportunity to grow up without the influence of social media.

## 6. Conclusions

Associations were found between the construction of body image and self-esteem for both groups and the knowledge provided by social media, which may or may not have good consequences. It is interesting to note that vegetarian women experience less pressure to look well, which may be related to the fact that they lead a lifestyle that is currently actively marketed online. Education is required on the connection between the use and internalization of social media knowledge and the development of healthy self-esteem and body evaluation.

## Figures and Tables

**Table 1 ijerph-20-02069-t001:** Sociodemographic status of groups (*n* = 460).

Sociodemographic Status		Group		*p*-Value
		Vegetarian (*n* = 200)	Traditional (*n* = 260)	
Education	Primary school	3 (1.5%)	3 (1.2%)	0.083
	High school	74 (36%)	92 (35%)	
	Vocational school	1 (0.5%)	2 (0.8%)	
	College	122 (62%)	163 (63%)	
Residence	Village	20 (10%)	63 (24.5%)	0.120
	City < 50,000. residents	32 (18%)	61 (23.5%)	
	City of 50–100 thousand. residents	25 (12%)	24 (9%)	
	City > 100,000. residents	123 (60%)	112 (43%)	
Marital status	Single	70 (36%)	72 (28%)	0.071
	Partnership	90 (44%)	103 (40%)	
	Marriage relationship	41 (20%)	85 (32%)	
Profession	Not working	50 (27%)	55 (21%)	0.065
	Casual work	20 (10%)	12 (5%)	
	Contract for example, work, commission	130 (63%)	193 (74%)	

**Table 2 ijerph-20-02069-t002:** Correlations between factors of social media use and selected lifestyle variables (*n* = 460).

Variable	Age	BMI	IP	IA	II	FI	SA	WC	PA	SE
Age	1	0.282 *	−0.252 *	−0.030	−0.221 *	−0.221 *	0.114	−0.021	0.07	0.103
Body Mass Index (BMI)		1	0.002	−0.019	0.008	−0.098	−0.023	−0.393 *	−0.162 *	0.22
Internalization pressure (IP)			1	0.500 *	0.630 *	0.693 *	−0.274 *	−0.347 *	−0.213 *	−0.290 *
Internalization of athleticism (IA)				1	0.270 *	0.362 *	−0.064	−0.068	0.022	−0.136
Internalization of the information searched (II)					1	0.544 *	−0.219 *	−0.255 *	−0.237 *	−0.244 *
Frequency of information searched (FI)						1	−0.272 *	−0.253 *	−0.144 *	−0.200 *
Sexual attractiveness (SA)							1	0.647 *	0.664 *	0.421 *
Weight control (WC)								1	0.700 *	0.375 *
Focusing on physical activity (PA)									1	0.344 *
Self-esteem (SE)										1

* statistically significant result at *p* ≤ 0.05.

**Table 3 ijerph-20-02069-t003:** Differences between traditional and vegetarian groups between studied factors (*n* = 460).

Variable	Vegetarian (*n* = 200)			Traditional (*n* = 260)			Statistics	
	Average	SD	Median	Average	SD	Median	Z	*p*
Internalization pressure	24.4	12.1	23	29.8	12.9	28	2.6	0.009 *
Internalization of athleticism	10.8	4.7	11	11.0	4.7	11	0.8	0.584
Internalization of the information searched	15.1	6.1	15	16.8	6.2	17	2.7	0.006 *
Frequency of information searched	10.6	5.2	8	12.4	6.0	12	3.6	0.000 *
Sexual attractiveness	44.8	8.9	46	45.9	8.9	46	0.2	0.882
Weight control	31.7	9.7	32	30.3	9.3	31	−1.5	0.139
Focusing on physical activity	29.4	7.8	29	28.9	7.4	29.5	−0.3	0.734
Self-esteem	18.6	3.5	19	18.7	3.0	19	−0.5	0.598

* statistically significant result at *p* ≤ 0.05.

## Data Availability

Not applicable.

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
