# Peer review of "The Role of Social Media in Internalizing Body Knowledge—A Cross-Sectional Study among Women with Different Food Preferences"

_ijerph, 2023, doi:10.3390/ijerph20032069_

Round 1
Reviewer 1 Report
This article estimates the impact of social networks on how women with different food preferences internalize body image. It is an interesting work, well-structured, and easy to read.
I have a doubt concerning the questionnaire used. The authors stated that a standard questionnaire named SATAQ is used, but they describe a pilot test. Do the authors pilot test the Polish version of the questionnaire?
As a minor comment, avoid changing the starting point of the y-axis in the figures or make it explicit in the text.
Author Response
Dear Reviewer,
Thank you for any suggested changes to our typescript. We have applied them to the text and highlighted them in blue for easy tracking of changes.
As for the comment on the piloting, we piloted the entire survey, which consisted of self-reported questions on sociodemographic issues and three psychometric tools. Polish adaptations of the tools were used, of course, as we already indicated in the original version of the article.
Thank you for your suggestion regarding the figures. By the decision of another reviewer, we removed all the figures, because, as noted, they duplicated the results in the text and Table 3.
Thank you!
Reviewer 2 Report
Overall, an interesting research paper, here are some comments which i believe can further strengthen the paper:
_The discussion section feels more like an expansion of the literature review rather than comparing the results of the present study with results from previous studies. There's also the inclusion of a review on "the impacts of social media men" despite not having any results on this aspect in the present study (.... Due to poor questionnaire maneuverability in the male population, the study was abandoned in this group and focused only on the female population).
_More proofreading can be conducted.
Author Response
Dear Reviewer,
Thank you for any suggested changes to our typescript. We have applied them to the text and highlighted them in blue for easy tracking of changes.
The discussion of the work has been rewritten and enriched with new data.
Thank you!
Reviewer 3 Report
The manuscript entitled “The role of social media in internalizing body knowledge - a cross-sectional study among women with different food preferences” is the study with an aim to examine the relationship between factors related to the use and internalization of body image knowledge contained in social media and the factors shaping self-assessment and self-esteem in women following a plant-based diet.
Abstract:
According to the instructions abstract should follow the style of structured one but without headings, so please remove headings from the abstracts.
Line 57 – “n diets” –with this n did you
Line 123 - No significant statistical differences (p<0.05) were found between the participants in the study – according to the? Sociodemographic parameters stated in the table 1 (if so please express actual p value for each characteristic)?
Results:
“In women following a plant-based diet, statistically significant associations were observed between factors related to the use and internalization of body image information from social media and age, BMI, self-esteem, and attitude toward one's own body. “– These results are shown on Fig 1 – Fig 3 but on those figures, there is no highlighted the existence of significant differences. Considering these results, you have duplicating the data presented at those figures and in table 3 (PI, I, PII), so authors should decide on one way of presenting the data.
Table 3 – you are missing legend (as you have at table 2 – tables should be understandable standing alone)
Conclusion:
“For both genders, associations were shown regarding the influence of knowledge presented by social media on the formation of body image and self-esteem, which may not necessarily have positive effects.” – general conclusion that are not obtained in the study is not recommended to be placed as a conclusion, especially when you didn’t conduct research on both gender
Authors stated that their strength of the study is the large study group, but if you have less respondent that is needed for the power of the study you cannot have this quote. Especially when this was online survey during five months and in Poland you have around 19,5 million of the females.
Author Response
Dear Reviewer,
Thank you for any suggested changes to our typescript. We have applied them to the text and highlighted them in blue for easy tracking of changes.
1 We have removed the section titles in the abstract.
2. We have removed the erroneous "n diet" notation.
3. Provided significance levels in Table 1.
4. Corrected Tables 2 and 3. Removed unnecessary figures that repeated results. The readability of the tables was corrected.
5. Incorrectly worded conclusion was corrected - it was "groups", not "gender".
6. Removed from the study's strengths the entry about the large study group.
Thank you!
Round 2
Reviewer 3 Report
I would like to thank the authors for their efforts to improve the manuscript.
Most of the remarks and suggestions have been adopted within the manuscript. Some rearrangements have been made to the text. Data that were displayed twice was removed.
The paper is now somewhat clearer, without general statements and without leading to ambiguous conclusions.